# Monopolistic vs. Competitive Supply Chain Concerning Selection of the Platform Selling Mode in Three Power Structures

**Lixi Zhou [1], Tijun Fan [1], Jie Yang [2] and Lihao Zhang [2,\*]**

[1] School of Business, East China University of Science and Technology, Shanghai 200237, China
[2] Logistics Research Center, Shanghai Maritime University, Shanghai 201306, China
* Correspondence: leehowzhang@hotmail.com

**Abstract:** This paper studies the selection of selling modes in a monopolistic and a competitive supply chain circumstance, where each supply chain comprises a supplier and an e-platform. The e-platform usually acts as a product reseller or serves as an online marketplace. The former is referred to as a reselling mode where the order fulfillment cost is paid by the supplier, and the latter is named as an agency selling mode where the platform pays for the order fulfillment cost. Motivated by the industrial cases, three power structures are utilized to capture the veridical market pricing. We find that the platform and the supplier's selling mode strategies conflict in a great majority of cases, except for the region in which both the platform agency fee and the order fulfillment cost are moderate. The players can coordinate by Pareto improvement, and the improved result shows that the optimal selling modes are a reselling mode in the fierce competitive supply chain circumstance and agency selling mode in the monopolistic circumstance or the mild competitive circumstance. Surprisingly, adopting a reselling mode is not only a better choice than adopting an agency selling mode in the fierce competitive supply chain circumstance, but this makes the supply chain obtain more of a payoff than in the monopolistic circumstance. Furthermore, it is worth noting that each player choosing an agency selling mode will result in a "prisoner's dilemma" in the competitive supply chain circumstance, where both players can obtain more payoffs with a reselling mode. In addition, the willingness of platforms and suppliers to play the role of "reseller" is the strongest under the *ps* structure and the weakest in the *ss* structure.

**Keywords:** platform selling mode; competitive supply chain; power structure; supply chain management

## 1. Introduction

As internet technology has developed, online retailing has seen significant growth in the last two decades. On 3 February 2022, American e-commerce giant Amazon announced that its annual revenue in 2021 was USD 469.8 billion, compared with USD 34.2 billion in 2010 (https://companiesmarketcap.com/amazon/revenue/ (accessed on 6 August 2022)). Tmall, a popular e-commerce platform in China, produced a total transaction volume of CNY 540.3 billion at the Double 11 shopping carnival in 2021, 10,806 times that of Tmall when it first set up the shopping festival in 2009 (https://www.tellerreport.com/business/2021-11-11-double-11-transcripts-released--tmall-540-3-billion-yuan--jingdong-over-349-1-billion-yuan.HkKUaaacwF.html (accessed on 6 August 2022)). Such a booming platform economy has attracted the attention of numerous scholars which mainly focused on the selling mode selection of an e-platform [1–4]. Generally, the e-platform traditionally acts as a product reseller or serves as an online marketplace [2]. The former is referred to as a reselling mode that, with the e-platform used as a traditional reseller, wholesales products from the supplier and then sells to consumers. For example, Be&Cheery and Three Squirrels both adopt a reselling mode to sell their products on JD.com [5]. The latter is named as an agency selling mode that, with the e-platform as an intermediary, provides an online marketplace for suppliers to sell products directly to consumers and charges

a certain percentage of agency fees correspondingly. For example, JD.com launched an agency selling mode to serve Coca-Cola and Pepsi.

In practice, some e-platforms such as Amazon and JD.com allow suppliers to select selling modes [5,6]. Nevertheless, other phenomena suggest that platforms independently decide their selling modes [1,7]. For example, JD.com merely provides a reselling mode for the several products of AMD, Huawei, and Microsoft, and Amazon only provides an agency selling mode for Kingston and SanDisk [8]. Despite the prevalence of the platform economy, the extant literature provides little guidance on contradictions between suppliers and platforms in the selection of selling modes. It is also unclear when platforms should give up decision making for selling modes. Therefore, this study aims to fulfill these research questions by examining the selling mode selection from the decentralized and centralized perspectives.

Meanwhile, current competition between firms has gradually expanded into competition between supply chains [9]. A report by Deloitte Consulting, based on a survey of more than 200 large manufacturers and distributors in the United States and Canada, showed that various industries have shown the trend of competition between supply chains and even platform supply chains. For example, in the highly competitive household appliance industry, Electrolux and SUNING.COM formed a supply chain that can compete with other supply chains such as Amoisonic and Pinduoduo Inc. Moreover, the skin care industry is another typical example for competitive platform supply chains. For instance, the supply chain composed of China's famous skin care brand Pechoin and e-commerce platform RED is in fierce competition with the supply chain composed of the famous US skin care brand Kiehl's and the cross-border e-commerce platform Koala. Therefore, this study expands the selling mode selection into a circumstance of competitive platform supply chains and allows differentiated selling mode strategies between two chains.

Motivated by industry practices and a literature gap, we develop a game-theoretical model to investigate selling mode selection from the perspectives of each player and centralized and integrated supply chains. We also expand the monopoly supply chain circumstance of the previous literature to the competitive supply chain circumstance. Either two-echelon supply chain is composed of an upstream supplier and a downstream e-platform, and two selling modes—reselling mode and agency selling mode—are considered. To achieve better practical and theoretical value, our model is comprehensive enough that it considers three power structures in industry practices and compares them. Our research questions are the following: (1) Should the platform give up decision making on selling modes? If not, which mode can be adopted to solve the decision conflicts? (2) How do competitive supply chains effect the equilibrium outcomes compared to monopolistic supply chains? (3) How does the power structure affect the selling mode strategy, and is the platform with more leadership power more favorable?

Through our investigation, the optimal order quantities, retail prices, wholesale prices, and profits of the retailer and manufacturer in each selling mode strategy are achieved. We discuss both the platform and the supplier's selling mode strategy in monopolistic and competitive circumstances. Our study shows that the platform and the supplier's selling mode strategy depend on the order fulfillment cost and the platform fee and have conflict in a great majority of cases. We improve the result with the Pareto principle, and the improved result shows that players should adopt the agency selling mode in the monopolistic circumstance or the mild competitive circumstance and adopt the reselling mode in the fierce competitive circumstance. In addition, we compare the single supply chain's payoff between the monopolistic and competitive circumstances and derive that the single supply chain in a monopolistic circumstance often gains more profits than in a competitive circumstance, except for a specific case of adopting the reselling mode when the competitive circumstance is fierce. Lastly, we analyze the impact of the power structure on the players' selling mode strategy and the result of Pareto improvement in the monopolistic and competitive circumstances.

The main contributions of this work are as follows. First, to the best of our knowledge, this paper is the first to explore the selling mode strategy in competitive supply chains. Second, we explore the selling mode strategy from the perspectives of each player and centralized and integrated supply chain and find that the platform giving up the decision making of selling mode selection might be more profitable. This conclusion can be interpreted for some phenomenon as some platforms being willing to allow suppliers to select the selling mode. Third, we find that the power structure will also affect the player's selling mode strategies, such as the platform acting as a leader, are more inclined toward choosing the reselling mode in the monopolistic supply chain and are more liable to achieve identical selling mode strategies in the competitive supply chain. These findings have great significance to enterprise production and operational decision in platform supply chain practices.

This paper is organized as follows. We show the literature review and innovations of our work in Section 2. Section 3 describes the model's set up, the game, and the related definitions in detail. In Section 4, we separately describe each combination of supply chain selling modes under monopolistic and competitive supply chain circumstances, and the corresponding equilibrium results in three power structures. Section 5 analyzes the impact of the supply chain circumstance on the selling mode decision and the impact of power structures on the selling mode decision. We summarize the research findings and give future research directions in Section 6. All proof results are provided in the Supplementary Materials.

## 2. Literature Review

The literature relevant to our work can be grouped into three streams: selling mode selection, competitive supply chain, and power structure.

Our work first contributes to the growing literature on the selling mode selection between the supplier and the retail platform. Since Hagiu [10] proposed two distinct selling modes (namely reselling and marketplace), more and more research has been devoted to uncovering the trade-offs between the two selling modes [11–14]. Hagiu and Wright [15] showed that the preference for the reselling or marketplace mode depends on whether the supplier or platform has more important information related to optimal customization of marketing campaigns for each specific product. Additionally, some of the literature explores sales strategies from different perspectives and identifies key drivers of selling mode selection, such as double upstream disadvantages [16], the logistics service strategy [17], controllable lead time and variable demand [18] and data-driven marketing [19]. Different from the above research, we investigated the impact of competition in the supply chain on selling mode decisions. Some of the literature examines the impact of upstream or downstream competition on selling mode selection. Abhishek et al. [1] studied a setting with one supplier and two e-tailers and found that e-tailers prefer agency selling when e-channel sales negatively impact traditional channel demand. Tian et al. [2] showed that competition among upstream suppliers significantly moderates the traditional marketplace advantage and found that the interaction between the order fulfillment cost and upstream competition intensity moderates the selection of an optimal intermediary mode. Furthermore, Zennyo [3] investigated selling mode selection between a monopolistic platform and two competing suppliers with different underlying demand and showed that when product substitution is low enough, the platform offers low royalties to induce suppliers to adopt agency contracts. This paper differs from the above studies in three key respects. First, our study focuses on the impact of chain-to-chain competition on selling mode decision. By comparing the supply chain circumstances of monopoly and competition, we capture the influence of supply chain competition on the preferences of all parties in the supply chain and then supplement the current literature flow of selling mode decision making in supply chain management. Second, we examine how three power structures influence the selling mode selection under monopolistic and competitive supply chains, which complements the gaps in the existing literature. Third, our study shows that by

adjusting the selling mode, the supply chain can be better in a competitive circumstance than in a monopolistic circumstance.

Our work is also related to the literature on supply chain competition. The pioneering study in the field of competition was conducted by McGuire and Staelin [20], who focused on the product market competition between two competing supply chains and found that each manufacturer will vertically integrate into retailing for products with low substitutability. Otherwise, the manufacturer will sell to specialized retailers. Based on McGuire and Staelin [20], much of the existing literature studies competitive supply chains in terms of price, inventory, service, sustainability, etc. [9,21–27]. Different from this, several examples in the literature have studied competition in supply chains that are integrated (i.e., manufacturers sell their own products) and decentralized (i.e., manufacturers sell their products to the market through retailers). For example, Anderson and Bao [21] demonstrated that the benefits of a centralized or decentralized supply chain are related to the coefficient of variation of market shares. Zhao and Shi [28] showed that a decentralized supply chain performs better in fierce market competition. However, we focus on analyzing the reasonableness of individual decisions of supply chain members from the perspective of the whole supply chain system. More specifically, our work differs from the above papers in the following aspects. First, they ignored the impact of the power structure on the performance of a competitive supply chain, while our research finds that the power structure makes a significant difference in the performance of supply chains in competition. Second, the existing literature rarely focuses on the comparison of the impact of competitive and monopolistic supply chain circumstances on members' decision making. Finally, we further demonstrate that the competitive circumstance may perform better than the monopoly circumstance by adjusting selling mode decisions.

The final stream of the literature relevant to our study is the power structure. Research on power structures can be traced back to Choi [29], and since then, many scholars have investigated the impact of specific power structures on the decision making and profits of supply chain members, such as manufacturer-Stackelberg (*MS*), retailer-Stackelberg (*RS*), and vertical Nash (*VN*) structures [14,30,31]. Among these works, some focused on a single supply chain [32–34], and some considered a dual-channel supply chain existing upstream or downstream in competition [35–38]. Our study extends the literature on selling mode decisions in competitive supply chains under different power structures, where each chain needs to formulate decisions for reselling or agency selling. The work closest to this paper is that of Pu et al. [4] who studied the manufacturer to decide the selling mode with the e-tailer by reselling or agency selling and a pricing strategy based on three power structures. Unlike our work, Pu et al. [4] focused on a dual-channel supply chain, where a manufacturer sells products through a retailer and a e-tailer, whereas chain-to-chain competition is our focus. In addition, Pu et al. [4] explored sales decisions only from the manufacturer's perspective, whereas we analyze the consistency of strategy selection between platforms and suppliers and obtain the equilibrium strategy after Pareto improvement in the two chains. Furthermore, Pu et al. [4] showed that the selection of the selling mode is related to the commission rate and power structure. However, we demonstrate that in a monopolistic supply chain, the agency selling mode in a centralized supply chain is always the equilibrium decision for any power structure.

Our paper also relates to research on behavioral operation management. (We sincerely thank the anonymous review for the insightful comments.) In practice, managers often deviate from the traditional decision of maximizing expected profits, resulting in poor supply chain performance [39]. Some of the literature analyzed the behavior mechanism by constructing a model based on the prospect theory [40,41], while others explored how to develop decision information systems to realize decision optimization of supply chains [42,43]. Recently, some scholars began to discuss the behavior mechanism in supply chain management in combination with new technologies, such as artificial intelligence and the blockchain [44,45]. In addition, D'Urso et al. [46] evaluated how individuals consider and use decision support systems in the context of Newsvendor. Our research discusses the

deviation of overall and individual behavior in decision making. This study verifies that individual behavior in the supply chain always makes the whole supply chain perform poorly, but we find that Pareto improvement among members can achieve optimization of the whole supply chain.

In sum, this paper moves beyond the existing literature in several dimensions. Platform supply chains will also be in the circumstance of chain-to-chain competition, a phenomenon seen in many industries but not explicitly considered in the literature. This work investigates the selling mode decisions of all parties in the supply chain and explores how reasonable members' decisions are from the perspective of the overall system. This research finds that the prisoner's dilemma exists in the competitive supply chain circumstance, and Pareto improvement can realize optimization of the supply chain system. Furthermore, it studies the interaction between competition and the selling mode decision and shows that adjusting the selling mode can make a single supply chain or whole supply chain system in a competitive circumstance outperform a monopoly circumstance. Finally, this work comprehensively discusses how different power structures affect the selling mode decision and supply chain competition. We show that the power structure can cause changes in supply chain parties' preferences for the selling mode and that vertical Nash structures are most beneficial to the whole supply chain in a competitive circumstance if the competition is mild, which has not been shown before. For clear presentation, we provide a comparison with the studies discussed above in Table 1.

**Table 1.** Summary of the relevant literature and contributions.

| Reference | Selling Mode Selection | Competitive Supply Chain | | Power Structure | | |
|---|---|---|---|---|---|---|
| | Reselling or Agency Selling | Upstream Competition | Downstream Competition | SS | VN | PS |
| Hagiu and Wright (2015) [15] | ✓ | ✓ | | | | |
| Abhishek et al. (2016) [1] | ✓ | | ✓ | | | |
| Tian et al. (2018) [2] | ✓ | ✓ | | | | |
| Zennyo (2020) [3] | ✓ | ✓ | | | | |
| Wu et al. (2009) [25] | | ✓ | ✓ | ✓ | ✓ | |
| Wu and Mallik (2010) [26] | | ✓ | ✓ | | | |
| Anderson and Bao (2010) [21] | ✓ | ✓ | ✓ | | | |
| Zhao and Shi (2011) [28] | ✓ | ✓ | | | | |
| Du et al. (2018) [22] | | ✓ | ✓ | | | |
| Feng and Liu (2022) [23] | | ✓ | ✓ | | | |
| Cai et al. (2009) [30] | | | | ✓ | ✓ | ✓ |
| Luo et al. (2017) [37] | | ✓ | | ✓ | ✓ | ✓ |
| Pu et al. (2021) [4] | ✓ | | ✓ | ✓ | ✓ | ✓ |
| This Paper | ✓ | ✓ | ✓ | ✓ | ✓ | ✓ |

## 3. The Model

We considered two different supply chain circumstances: a monopolistic supply chain circumstance and a competitive supply chain circumstance. The former is only a two-level supply chain (labeled chain 1) composed of a supplier and a retail platform, while the latter describes two competitive supply chains (labeled chain 1 and chain 2) that produce substitutable products, where each two-level supply chain consists of a supplier and a retail platform.

The supplier sells products on the retail platform through the reselling mode or agency selling mode. In the reselling mode, the platform, as a reseller, wholesales products from the supplier and resells the products to consumers. Therefore, the pricing power of products in the retail channel is controlled by the platform. Note that in the era of electronic commerce development, there is a cost for the company that cannot be ignored. Following the literature [47,48], order fulfillment (that is, the successful delivery of products to consumers) is the most critical and costly link in online sales. In order to complete the order, the company needs to bear costs such as warehouse construction or leasing costs,

package manual processing fees, and delivering goods to consumers. Since the cost of storing and recruiting staff is considerable, it can be regarded as a fixed cost. At the same time, the delivery cost of online shopping is usually borne by the consumers. Therefore, the order fulfillment cost is usually assumed to be a fixed cost [2], which we use *F* to represent. In this mode, order fulfillment is borne by the platform.

In the agency selling mode, the platform acts as an intermediary to allow the supplier to sell products directly to consumers on its website and charges the platform's agency fee $\alpha$. The platform transfers the pricing power of products to the supplier, so the supplier bears the cost of order fulfillment. In order to avoid some trivialities, we assume that the order fulfillment costs borne by the platform and supplier are equal [2]. The agency fee is usually considered a fixed proportion of the fee, which has different proportions according to different product categories. For example, JD.com, one of the largest e-commerce retailers in China, has agency rates ranging from 5% to 12% for most of its products.

### 3.1. Model Framework

Motivated by the industrial cases, this paper considers two supply chain circumstances: a monopolistic supply chain circumstance and a competitive supply chain circumstance. In either circumstance, both the supplier and the platform are faced with the selection of the selling mode, leading to the following scenarios:

***The monopolistic supply chain circumstance***: In this case, the supply chain consists of a supplier and a platform. Thus, there are two scenarios that will be investigated according to the players' selling mode strategies: (1) scenario *R*, where supplier 1 or platform 1 chooses the reselling mode, and (2) scenario *A*, where supplier 1 or platform 1 chooses the agency selling mode. Figure 1 describes the supply chain structure under the monopolistic supply chain circumstance.

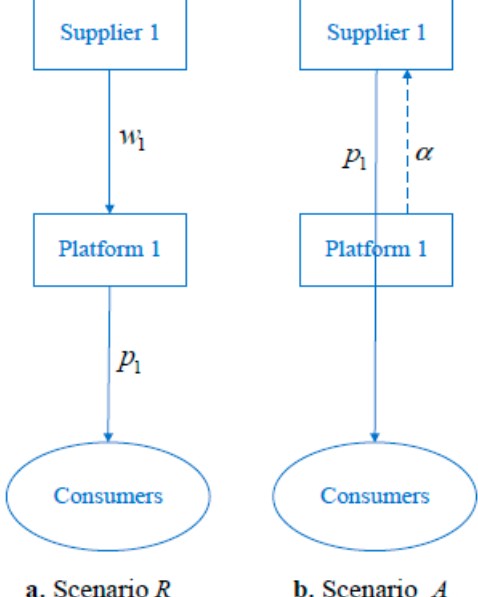

**Figure 1.** The supply chain structure of the monopolistic supply chain. (**a**) Scenario *R*; (**b**) Scenario *A*.

***The competitive supply chain circumstance:*** In this case, there are two competing supply chains in the market, and each chain needs to decide which selling mode to adopt. Therefore, there are three different scenarios: (1) scenario *RR*, where both chains choose the reselling mode, (2) scenario *RA*, where one supply chain (chain 1) chooses the reselling mode and the other (chain 2) chooses the agency selling mode and where we assume that two competitive supply chains are symmetric, leading us to investigate one scenario in the hybrid mode, and (3) scenario *AA*, where the supplier or the platform chooses the agency selling mode. Without losing generality, we assume that the two chains have equal order

fulfillment costs and agency rates. Figure 2 describes the supply chain structure under the competitive supply chain circumstance.

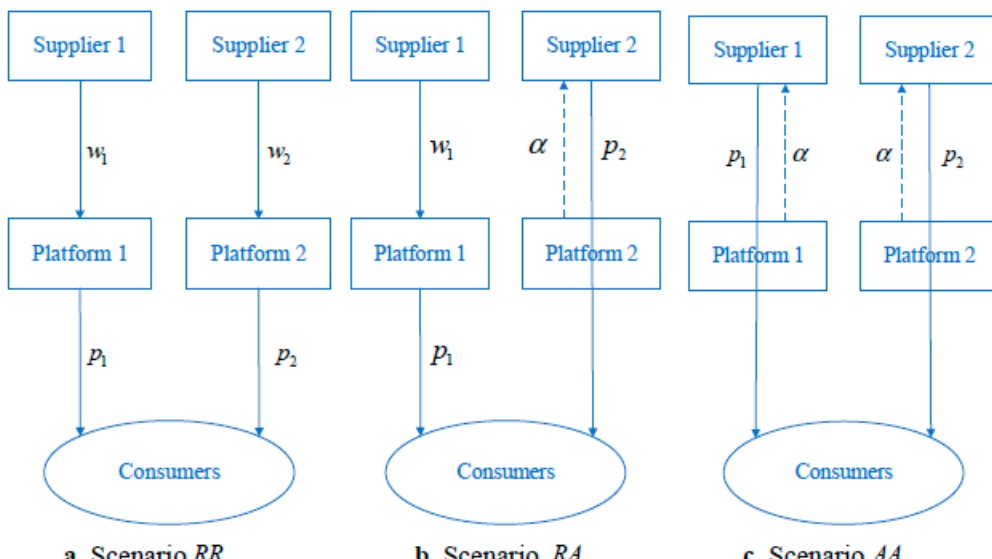

**Figure 2.** The supply chain structure of the competitive supply chain. (**a**) Scenario *RR*; (**b**) Scenario *RA*; (**c**) Scenario *AA*.

### 3.2. Demand Function

We assume that the two supply chains are symmetric and differentiated. Following the literature on competitive supply chains and supply chain management [49,50], we define the demand competition for competitive supply chains by adopting a utility function for a representative consumer:

$$U = \sum_{i=1,2} \left(aq_i - \frac{q_i^2}{2}\right) - dq_1 q_2 - \sum_{i=1,2} p_i q_i \tag{1}$$

where $q_i$ denotes the consumer demands of chain $i$'s retail channel, $p_i$ refers to the market prices for chain $i$'s retail channel, $a$ is the market size, and $d$ ($0 < d < 1$) represents the competition intensity between two channels, where a larger $d$ indicates a higher degree of channel competition. The above utility function in Equation (1) induces a linear demand structure. Maximization of Equation (1) yields the demand for chain $i$'s retail channel in the competitive supply chain as follows:

$$\begin{aligned} q_1(p_1, p_2) &= \frac{(1-d)a - p_1 + dp_2}{1 - d^2} \\ q_2(p_1, p_2) &= \frac{(1-d)a - p_2 + dp_1}{1 - d^2} \end{aligned} \tag{2}$$

Under the monopolistic supply chain circumstance, the demand of chain $2'$s retail channel will be zero (i.e., $q_2 = 0$). Maximization of Equation (1) yields the demand for chain $1'$s retail channel as $q_1 = a - p_1$.

### 3.3. Timeline of the Game

At the beginning of production season, the supplier $i$ provides products, and then the chain $i$ formulates the selling mode decision. In the selling season, the supply chain members' operation strategies in the different power structure are as follows:

***Platform-Stackelberg (ps) structure***: In scenario $R$ or $RR$, platforms 1 and 2 (if any) simultaneously set the margin profit $m_i^{ps,y}$ ($y \in \{R, A, RR, RA, AA\}$), and then suppliers 1 and 2 (if any) simultaneously decide the wholesale price $w_i^{ps,y}$. In scenario $A$ or $AA$, platforms 1 and 2 (if any) decide the retail price $p_j^{ps,y}$. In scenario $RA$, platform 1 sets the

margin profit $m_1^{ps,y}$. Then, supplier 1 decides the wholesale price $w_1^{ps,y}$, and platform 2 (if any) decides the retail price $p_2^{ps,y}$.

***Vertical Nash (vn) structure***: In scenario *R* or *RR*, suppliers 1 and 2 (if any) simultaneously decide the wholesale price $w_i^{vn,y}$, and then platforms 1 and 2 (if any) set the margin profit $m_i^{vn,y}$. In scenario *A* or *AA*, platforms 1 and 2 (if any) decide the retail price $p_i^{vn,y}$. In scenario *RA*, supplier 1 decides the wholesale price $w_1^{vn,y}$. Then, platform 1 sets the margin profit $m_1^{vn,y}$, and platform 2 (if any) decides the retail price $p_2^{vn,y}$.

***Supplier-Stackelberg (ss) structure***: In scenario *R* or *RR*, suppliers 1 and 2 (if any) simultaneously decide the wholesale price $w_i^{ss,y}$, and then platforms 1 and 2 (if any) set the retail price $p_i^{ss,y}$. In scenario *A* or *AA*, platforms 1 and 2 (if any) decide the retail price $p_i^{ss,y}$. In scenario *RA*, supplier 1 decides the wholesale price $w_1^{ss,y}$. Then, platform 1 sets the retail price $p_1^{ss,y}$, and platform 2 (if any) decides the retail price $p_2^{ss,y}$.

### 3.4. Model Assumptions and Notation Definitions

For clarity, we summarize all assumptions as follows, and the parameters and variables used in this paper are summarized in Table 2:

**Assumption 1.** *$d < 0.664$. This states that the order quantities are non-negative (i.e., $q_i^{x,y} > 0$) and the players should be profitable (i.e., $p_i^{x,y} > w_i^{x,y}$) in the potential market [51–53].*

**Assumption 2.** *The order fulfillment F is a fixed cost, and we assume that the order fulfillment costs borne by the platform and supplier are equal [2].*

**Assumption 3.** *The agency fee α is usually considered to be a fixed proportion of the fee, which has different proportions according to different product categories.*

**Assumption 4.** *Under the competitive supply chain circumstance, two chains produce substitutable products and are symmetric.*

**Assumption 5.** *The suppliers and the platforms are both risk-neutral, and this information is completely symmetrical.*

**Table 2.** Notations and explanations.

| Notation | Explanation |
|---|---|
| $a$ | Market size. |
| $d$ | Competition intensity of the two supply chains. |
| $\alpha$ | The platform fee rate. |
| $F_i$ | The order fulfillment cost, where $i \in \{1, 2\}$. |
| $q_i^{x,y}$ | The customer demand of supply chain $i$ in scenario $y$ under $x$ power structure, where $x \in \{ps, ss, vn\}$ and $y \in \{R, A, RR, RA, AA\}$. |
| $p_i^{x,y}$ | The sales price of supply chain $i$ in scenario $y$ under $x$ power structure. |
| $m_i^{x,y}$ | The platform's marginal profit with supply chain $i$ in scenario $y$ under $x$ power structure. |
| $w_i^{x,y}$ | The supplier's wholesale price with supply chain $i$ in scenario $y$ under $x$ power structure. |
| $\pi_{p,i}^{x,y}$ | The platform's profit of supply chain $i$ in scenario $y$ under $x$ power structure. |
| $\pi_{s,i}^{x,y}$ | The supplier's profit with supply chain $i$ in scenario $y$ under $x$ power structure. |

## 4. Equilibrium Analysis

In practice, for a product category, the supply chain is monopolized before the competitive supply chain circumstance is formed. For example, in the electrical industry, after JD.com established an online mall in 2003 and cooperated with the supplier, SUNING.COM, which formally opened an e-commerce platform to serve its supplier in 2009. In the beauty and skin care industry, RED began to operate 3 years later than Jumei. Based on the above background, in order to comprehensively explore the impact of the competitive circumstance on the selling mode decision making in the platform supply chain, we also discuss the two different market circumstances of monopolistic and competitive and perform a comparative analysis. Section 4.1 shows the selling mode decisions of supply chain members in a monopolistic circumstance, while Section 4.2 describes the equilibrium decisions of two supply chains in a competitive circumstance.

### 4.1. The Monopolistic Supply Chain

In this section, we discuss the optimal selling mode strategy of a supply chain under a monopolistic supply chain circumstance. There are two scenarios that will be investigated: (1) scenario *R* (the chain opts for the reselling mode) and (2) scenario A (the chain opts for the agency selling mode). First, by comparing the optimal outcomes of supplier 1 and platform 1 under different scenarios, we obtain the optimal strategies of the players. Based on the above results, we discuss whether there is conflict in the strategic choice of each player and discuss the optimal selling mode strategy after Pareto improvement in the supply chain.

#### 4.1.1. Scenario R: Reselling Mode

In this scenario, the supply chain opts for the reselling mode (i.e., *R*); that is, platform 1, as a reseller, wholesales products from supplier 1 at wholesale prices and decides the retail price, and platform 1 bears the cost of order fulfillment. Therefore, the objective functions of supplier 1 and platform 1 are given by

$$\begin{cases} \pi_{s,1}^{x,R} = w_1^{x,R} q_1^{x,R} \\ \pi_{p,1}^{x,R} = \left(p_1^{x,R} - w_1^{x,R}\right) q_1^{x,R} - F_1 \end{cases} \tag{3}$$

where superscript $x$ indicates three different power structures, namely the platform-Stackelberg structure (*ps*), vertical Nash structure (*vn*), and supplier-Stackelberg structure (*ss*), and $\pi_{s,1}^{x,R}$ and $\pi_{p,1}^{x,R}$ are the expected profit of supplier 1 and platform 1, respectively. By using backward induction, we can obtain the optimal solution. In the *ps* structure, we first obtain the equilibrium wholesale price by maximizing the profit of the supplier 1. Then, we derive the optimal margin profit by maximizing the profit of platform 1. In the *vn* structure, the margin profit can be obtained by maximizing the profit of platform 1. Then, we derive the optimal wholesale price by solving $\partial \pi_{s,1}^{x,R} / \partial w_1^{x,R} = 0$. In the *ss* structure, we first obtain the equilibrium retail price by maximizing the profit of supplier 1. Then, we derive the optimal wholesale price by maximizing the profit of supplier 1. Table 2 summarizes the optimal results for scenario *R* under the three different power structures.

#### 4.1.2. Scenario A: Agency Selling Mode

In this scenario, the supply chain opts for the agency selling mode (i.e., *A*). At this time, platform 1, as an intermediary, charges supplier 1 the agency fee $\alpha$ for selling products, and the supplier bears the cost of order fulfillment. Therefore, the objective functions of supplier 1 and platform 1 are given by

$$\begin{cases} \pi_{s,1}^{x,A} = (1-\alpha) p_1^{x,A} q_1^{x,A} - F \\ \pi_{p,1}^{x,A} = \alpha p_1^{x,A} q_1^{x,A} \end{cases} \tag{4}$$

With the second-order derivative $\partial^2 \pi_{s,1}^{x,A} / \partial (p_{s,1}^{x,A})^2 < 0$, we find that the optimum solution of $p_1^{x,A}$ is determined by $\partial \pi_{s,1}^{x,A} / \partial p_{s,1}^{x,A} = 0$. Then, we can obtain the optimal profits in scenario $A$ as shown in the Table S1 of Supplementary Materials.

### 4.1.3. Equilibrium Strategy of the Selling Mode

With the players' optimal profits, the players' selling mode decision equilibrium under the monopolistic supply chain circumstance is derived as shown in the following proposition:

**Proposition 1.** *The equilibrium of the players' selling mode decisions under the monopolistic supply chain circumstance in any power structure are as follows:*

*(i) For platform 1, the reselling mode (i.e., R) is the optimal decision if $F \in (0, \overline{F}_{p1}^x]$ and the agency selling mode (i.e., A) otherwise;*
*(ii) For supplier 1, the agency selling mode (i.e., A) is the optimal decision if $F \in (0, \overline{F}_{s1}^x]$ and the reselling mode (i.e., R) otherwise;*
*(iii) Both platform 1 and supplier1 prefer the agency selling mode if $F \in (\overline{F}_{p1}^x, \overline{F}_{s1}^x]$.*
*where $x \in \{ps, vn, ss\}$, $\overline{F}_{p1}^{ps} = (1 - 2\alpha)a^2/8$, $\overline{F}_{s1}^{ps} = (3 - 4\alpha)a^2/16$, $\overline{F}_{p1}^{vn} = (4 - 9\alpha)a^2/36$, $\overline{F}_{s1}^{vn} = (5 - 9\alpha)a^2/36$, $\overline{F}_{p1}^{ss} = (1 - 4\alpha)a^2/16$, and $\overline{F}_{s1}^{ss} = (1 - 2\alpha)a^2/8$.*

Proposition 1 (i) demonstrates platform 1's selling mode decision equilibrium. It is noteworthy that the order fulfillment cost is a decisive factor for platform 1's choice of the selling mode. Obviously, platform 1 chooses the reselling mode if and only if the order fulfillment cost is below a certain threshold. Conversely, if the order fulfillment cost is above the threshold, the supplier will prefer the agency selling mode. As an example of the *ps* structure in Figure 3, when the order fulfillment cost is lower than $\overline{F}_{p1}^{ps}$, platform 1 will execute the reselling mode (the yellow region). Platform 1 chooses the agency selling mode otherwise (the white region and the blue region). The reason for this is that platform 1, as the reseller, must bear a high order fulfillment cost in the reselling mode, but in the agency selling mode, platform 1 does not play a "reseller" role and thereby does not have to pay the order fulfillment cost. When the order fulfillment cost is expensive, for platform 1, the agency selling mode will be more profitable than the reselling mode.

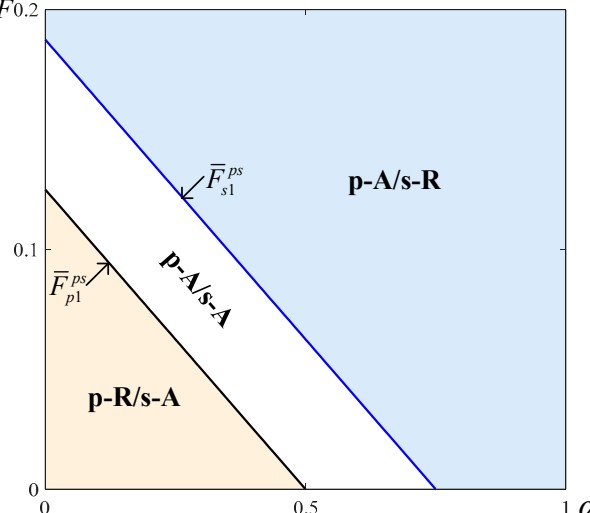

**Figure 3.** The players' selling mode decision under the monopolistic supply chain circumstance in the *ps* structure ($a = 1$). (i) The three power structures have similar results. Hence, we only take the *ps* structure as an example to show proposition 1. (ii) $j - y$ means player $j$ prefers scenario $y$, where $j \in \{p, s\}$ and $y \in \{R, A, RR, RA, AA\}$.

Next, Proposition 1 (ii) explores supplier 1's selling mode decision equilibrium. It is not difficult to find that supplier 1's selling mode decision is also related to the order fulfillment cost. Unlike platform 1's choice, supplier 1 prefers the agency selling mode if and only if the order fulfillment cost is below a certain threshold. Once the order fulfillment cost is above the threshold, supplier 1 will adopt the reselling mode. Figure 3 shows an example under the *ps* structure. We found that supplier 1 prefers the agency selling mode when the order fulfillment cost is lower than $\overline{F}_{s1}^{ps}$, as shown in the yellow region and the white region. Supplier 1 chooses the reselling mode otherwise, as shown in the blue region. This occurs because supplier 1, as the reseller in the agency selling mode rather than the reselling mode, must bear the order fulfillment cost in the agency selling mode.

Then, part (iii) of Proposition 1 investigates the players' selling mode decision equilibrium. It is noteworthy that platform 1 and supplier 1 can achieve a consistent equilibrium result where they both execute the agency selling mode when order fulfillment costs are moderate. Apart from that, when the order fulfillment cost is too low or too expensive, platform 1 and supplier 1 cannot achieve an equilibrium outcome. For the example in Figure 3, the white region highlights that both players choose the agency selling mode. When the order fulfillment cost exceeds the threshold $\overline{F}_{s1}^{ps}$, supplier 1 prefers the reselling mode and is thereby unable to achieve an equilibrium outcome, shown in the blue region. When the order fulfillment cost is below the threshold $\overline{F}_{p1}^{ps}$, platform 1 prefers the reselling mode and is thereby unable to achieve an equilibrium outcome, shown in the yellow region.

Because the optimal strategy of supplier 1 and platform 1 always makes it difficult to reach an agreement in a great majority of cases, we implement Pareto improvement on the profit of players and define it as a centralized supply chain system under the monopolistic supply chain circumstance. Then, we discuss the optimal strategy of the centralized supply chain system. The corresponding results are shown in Proposition 2:

**Proposition 2.** *Under the monopolistic supply chain circumstance, for the centralized supply chain, the agency selling mode (i.e., A) is always the equilibrium decision in any power structure.*

Proposition 2 reflects that for any power structure, the agency selling mode is the optimal choice for the total supply chain system. In other words, supplier 1 and platform 1 can always achieve optimization by Pareto improvement. Recalling proposition 1, when the order fulfillment cost is low, platform 1 always prefers the reselling mode, while supplier 1 prefers the agency selling mode. At this time, supplier 1 can promote the platform to accept the agency selling mode by transferring part of the revenue to platform 1 so that both players can achieve the optimal profit. On the contrary, if the order fulfillment cost is high, then supplier 1 is more willing to opt for the reselling mode, while platform 1 is inclined to choose the agency selling mode. In this way, platform 1 can encourage supplier 1 to prefer the agency selling mode by transferring part of the revenue to supplier 1 so that both players obtain the optimal profit. Note that the agency selling mode is always the optimal strategy, which may be because supplier 1 and platform 1 can achieve Pareto improvement through the agency rate in the agency selling mode. This conclusion also explains why, in business practice, the platform and supplier gradually prefer to adopt the agency selling mode for cooperation.

### 4.2. The Competitive Supply Chain

In this section, we investigate a competitive supply chain circumstance which has two supply chains. Therefore, the following four scenarios are discussed: (1) *RR*, where both chains opt for the reselling mode, (2) *RA* and (3) *AR*, where the two chains opt for the opposite selling mode, and (4) *AA*, where both chains opt for the agency selling mode. We derive the optimal solutions under different scenarios and capture the selling mode equilibrium strategy of the supply chain members and supply chain systems.

### 4.2.1. Scenario RR: Either Supply Chain Chooses the Reselling Mode

In this scenario, each chain opts for the reselling mode (i.e., *RR*); that is, the two platforms, as the resellers, wholesale products from the two suppliers at wholesale prices and decide the retail prices. At this time, the order fulfillment costs are borne by the two platforms. Therefore, the objective functions of the players are given by

$$
\begin{cases}
\pi_{s,j}^{x,RR} = w_j^{x,RR} q_j^{x,RR} \\
\pi_{p,j}^{x,RR} = (p_j^{x,RR} - w_j^{x,RR}) q_j^{x,RR} - F
\end{cases}
\tag{5}
$$

where $\pi_{s,j}^{x,RR}$ and $\pi_{p,j}^{x,RR}$ represent the expected profit of the two suppliers and two platforms, respectively, and $j = 1, 2$. Via backward induction, we can find the optimal solution under scenario *RR*. In the *ps* structure, in the second stage, we derive the equilibrium wholesale prices by maximizing the profit of the suppliers, and in the first stage, the platforms determine the optimal margin profits based on the suppliers' wholesale prices. In the *vn* structure, we derive the equilibrium margin profits by maximizing the profit of the platforms, and then the suppliers determine the optimal wholesale prices based on the platforms' margin profits. In the *ss* structure, we obtain the equilibrium retail prices by maximizing the profit of the platforms, and in the first stage, the suppliers determine the optimal wholesale prices based on the platforms' retail prices. We summarize the outcomes of scenario *RR* in the Table S2 of Supplementary Materials.

### 4.2.2. Scenario RA (or AR): Only One Supply Chain Chooses the Reselling Mode

In this scenario, one chain (chain 1) chooses the reselling mode, and the other chain (chain 2) chooses the agency selling mode (i.e., *RA*). Therefore, platform 1 wholesales products from supplier 1 and decides the retail price. However, platform 2, as the intermediary, charges supplier 2 the agency fee $\alpha$ for selling the products. At this time, the order fulfillment costs are borne by platform 1 and supplier 2. Therefore, the objective functions of the players in the two chains are given by

$$
\begin{cases}
\pi_{s,1}^{x,RA} = w_1^{x,RA} q_1^{x,RA} \\
\pi_{p,1}^{x,RA} = (p_1^{x,RA} - w_1^{x,RA}) q_1^{x,RA} - F
\end{cases}
\tag{6}
$$

$$
\begin{cases}
\pi_{s,2}^{x,RA} = (1 - \alpha) p_2^{x,RA} q_2^{x,RA} - F \\
\pi_{p,2}^{x,RA} = \alpha p_2^{x,RA} q_2^{x,RA}
\end{cases}
\tag{7}
$$

We solve this game using backward induction. In the *ps* structure, in the second stage, we find the equilibrium wholesale price $w_1^{ps,RA}$ and retail price $p_2^{ps,RA}$ by maximizing the profits of supplier 1 and supplier 2, respectively, and in the first stage, platform 1 determines the optimal margin profits $m_1^{ps,RA*}$ based on supplier 1's wholesale prices. In the *vn* structure, in the second stage, we find the equilibrium margin profits $m_1^{ps,RA}$ and retail price $p_2^{ps,RA}$ by maximizing the profits of platform 1 and supplier 2, respectively, and in the first stage, supplier 1 determines the optimal wholesale price $w_1^{ps,RA*}$ based on platform 1's margin profits. In the *ss* structure, we find the equilibrium retail price $p_1^{ps,RA}$ and $p_2^{ps,RA}$ by maximizing the profits of platform 1 and supplier 2, respectively, and in the first stage, platform 1 determines the optimal wholesale price $w_1^{ps,RA}$ based on platform 1's retail price. We summarize the outcomes of scenario *RA* in the Table S3 of Supplementary Materials.

### 4.2.3. Scenario AA: Either Supply Chain Chooses the Agency Selling Mode

In this scenario, each chain opts for the agency selling mode (i.e., *AA*); that is, the two platforms, as the intermediaries, charge suppliers the agency fee $\alpha$ for selling products. At

this time, the order fulfillment costs are borne by the two suppliers. Therefore, the objective functions of the players are given by

$$
\begin{cases}
\pi_{s,j}^{x,AA} = (1-\alpha)p_j^{x,AA}q_j^{x,AA} - F \\
\pi_{p,j}^{x,AA} = \alpha p_j^{x,AA}q_j^{x,AA}
\end{cases}
\tag{8}
$$

With the second-order derivative $\partial^2 \pi_{s,j}^{x,AA}/\partial(p_{s,j}^{x,AA})^2 < 0$, we can obtain the suppliers' optimal retail prices to maximize their payoffs. We summarize the outcomes of scenario *AA* in the Table S4 of Supplementary Materials.

4.2.4. Equilibrium Strategy of the Selling Mode

In this subsection, we first find the selling mode equilibrium strategies of two platforms and two suppliers by comparing the profits of supply chain members under different scenarios. Furthermore, we analyze the consistency of strategy selection between the platforms and suppliers and obtain the equilibrium strategy after Pareto improvement in the two chains. On this basis, we examine whether there is a prisoner's dilemma phenomenon in the two chains and discuss the selling mode equilibrium strategy of the whole supply chain system after Pareto improvement.

First, by comparing the optimal profits of the two platforms under four different scenarios, we obtain the selling mode equilibrium strategy of the platforms, as shown in Proposition 3:

**Proposition 3.** *Under the competitive supply chain circumstance, the equilibrium of the platforms' selling mode decision in any power structure is expressed as follows:*

*(i) When $\alpha \in (0, \overline{\alpha}_p^x]$, both platforms prefer the reselling mode (i.e., RR) as the equilibrium decision if $F \in (0, \overline{F}_{p2}^x]$, and both platforms prefer the agency selling mode (i.e., AA) as the equilibrium decision if $F \in (\overline{F}_{p2}^x, +\infty)$;*
*(ii) When $\alpha \in (\overline{\alpha}_p^x, 1)$, both platforms prefer the reselling mode (i.e., RR) as the optimal equilibrium if $F \in (0, \overline{F}_{p2}^x]$, both platforms prefer the opposite selling mode (i.e., RA or AR) as the equilibrium decision if $F \in (\overline{F}_{p2}^x, \overline{F}_{p3}^x]$, and both platforms prefer the agency selling mode (i.e., AA) as the equilibrium decision if $F \in (\overline{F}_{p3}^x, +\infty)$, where $x \in \{ps, vn, ss\}$. The expressions of $\overline{F}_{p2}^x$ and $\overline{F}_{p3}^x$ are shown in the Supplementary Materials.*

Proposition 3 shows that under the competitive supply chain circumstance, the selling mode equilibrium decision of the platforms is jointly affected by the agency rate and the order fulfillment cost in any power structure, as shown in Figure 4. When the agency rate is low (i.e., $\alpha \in (0, \overline{\alpha}_p^x]$), the lower order fulfillment cost (i.e., $F \in (0, \overline{F}_{p2}^x]$) will encourage both platforms to choose the reselling mode, as shown in the yellow region, and the higher order fulfillment cost (i.e., $F \in (\overline{F}_{p2}^x, +\infty)$) will make the platforms prefer the agency selling mode, as shown in the blue region. This conclusion is in line with Proposition 1 (i). Obviously, in the reselling mode, the cost of order fulfillment is borne by the platforms. Thus, higher order fulfillment costs bring more expenses to platforms that choose the reselling mode, resulting in lower profits than the agency selling mode. However, it is worth noting that when the agency rate is high (i.e., $\alpha \in (\overline{\alpha}_p^x, 1)$), we find that the selling mode equilibrium strategies of the two platforms are not always consistent. Specifically, when the order fulfillment cost is moderate (i.e., $F \in (\overline{F}_{p2}^x, \overline{F}_{p3}^x]$), the two platforms will always choose the opposite selling mode (i.e., scenario *RA* or *AR* is the equilibrium), as shown in the pink region. This is because, intuitively, with the increase in the agency rate, the platform is more willing to choose the agency selling mode. However, the low order fulfillment cost also encourages them to choose the reselling mode. Therefore, for the two platforms, it is an equilibrium strategy for one to choose the reselling mode and the other to choose the agency selling mode. In addition, below the pink area (i.e., when the order fulfillment cost is low),

the two platforms prefer to choose the reselling mode, while above the pink area (i.e., when the order fulfillment cost is high), the two platforms prefer the agency selling mode.

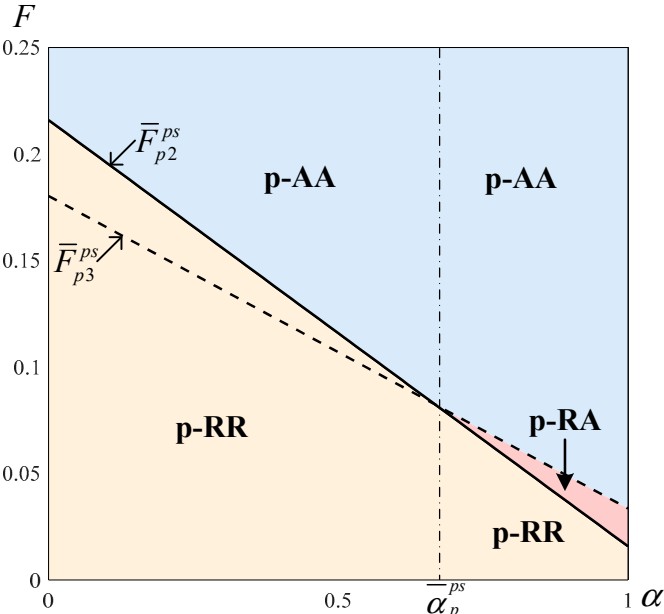

**Figure 4.** The platforms' selling mode decisions under the competitive supply chain circumstance in the *ps* structure ($a = 1$, $d = 0.45$).

Next, we further discuss the selling mode equilibrium strategy of two suppliers in the supply chain. All the results are shown in Lemma 1:

**Lemma 1.** *Under the competitive supply chain circumstance, the equilibrium of the suppliers' selling mode decisions in any power structure is expressed as follows:*

*(i) When $\alpha \in (0, \overline{\alpha}_s^x]$, both suppliers prefer the agency selling mode (i.e., AA) as the equilibrium decision if $F \in (0, \overline{F}_{s3}^x]$, both suppliers prefer the opposite selling mode (i.e., RA or AR) if $F \in (\overline{F}_{s3}^x, \overline{F}_{s2}^x]$, and both suppliers prefer the reselling mode (i.e., RR) as the equilibrium decision if $F \in (\overline{F}_{s2}^x, +\infty)$;*
*(ii) When $\alpha \in (\overline{\alpha}_s^x, 1)$, both suppliers prefer the agency selling mode (i.e., AA) as the equilibrium decision if $F \in (0, \overline{F}_{s2}^x]$, and both suppliers prefer the reselling mode (i.e., RR) as the equilibrium decision if $F \in (\overline{F}_{s2}^x, +\infty)$, where $x \in \{ps, vn, ss\}$. The expressions of $\overline{F}_{s2}^x$ and $\overline{F}_{s3}^x$ are shown in the Supplementary Materials.*

Similar to Lemma 1, in the competitive supply chain circumstance, the agency rate and order fulfillment cost jointly affect the suppliers' selling mode equilibrium decisions for any power structure, as shown in Figure 5. When the agency rate is low (i.e., $\alpha \in (0, \overline{\alpha}_s^x]$), the two suppliers can bear the fulfillment cost and prefer to choose the agency selling mode if the order fulfillment cost is low (i.e., $F \in (0, \overline{F}_{s3}^x]$), as shown in the blue region. Recalling Proposition 1 (ii), lower order fulfillment costs encourage supplier 1 to choose the agency selling mode. Therefore, in order to maintain high profits, both suppliers prefer the agency selling mode. Moreover, if the order fulfillment cost is moderate (i.e., $F \in (\overline{F}_{s3}^x, \overline{F}_{s2}^x]$), then the two suppliers always choose the opposite selling modes; that is, one supplier prefers the reselling mode, and the other supplier prefers the agency selling mode, as shown in the pink region. The reason for this is similar to what Proposition 3 explains, so it will not be repeated here. In addition, if the order fulfillment cost is high (i.e., $F \in (\overline{F}_{s2}^x, +\infty)$), then the two suppliers are always more willing to opt for the reselling mode to avoid bearing high costs, as shown in the yellow region. Different from the above conclusion, two suppliers will always choose the same selling mode if the agency rate is high (i.e., $\alpha \in (\overline{\alpha}_s^x, 1)$), which

is also different from the platforms' equilibrium strategy. In this case, if the order fulfillment cost is low (i.e., $F \in (0, \overline{F}_{s2}^x])$, then the two suppliers always prefer the agency selling mode; otherwise, the reselling mode is an equilibrium strategy.

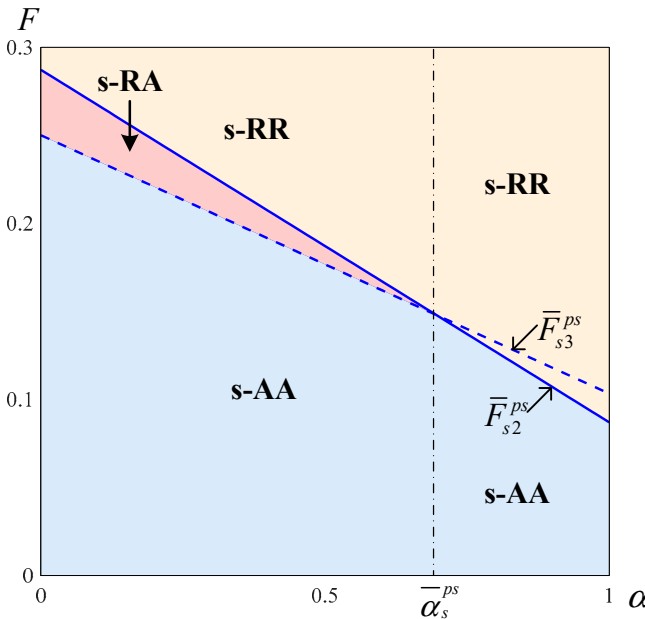

**Figure 5.** The suppliers' selling format decisions under the competitive supply chain circumstance in the *ps* structure ($a = 1$, $d = 0.45$).

Intuitively, the selling mode equilibrium strategies of the platforms and suppliers are not identical. Therefore, we next discuss whether there is consistency in the strategic choices of players in each supply chain:

**Proposition 4.** *Under the competitive supply chain circumstance, the equilibrium of the suppliers' and the platforms' selling mode decisions in any power structure is expessed as follows:*

*(i) Each player prefers the agency selling mode (i.e., AA) as the equilibrium decision in the regions* $\Omega_{AA}^x$;

*(ii) Each player prefers the opposite selling mode (i.e., RA or AR) as the equilibrium decision in the regions* $\Omega_{RA}^x$;

*(iii) Each player prefers the reselling mode (i.e., RR) as the equilibrium decision in the regions* $\Omega_{RR}^x$,

*where* $x \in \{ps, vn, ss\}$, $\Omega_{AA}^{ps} \equiv \left\{ d \in (0, 0.543], F \in (max\{\overline{F}_{p2}^{ps}, \overline{F}_{p3}^{ps}\}, min\{\overline{F}_{s2}^{ps}, \overline{F}_{s3}^{ps}\}] \right\}$,

$\Omega_{AA}^{vn} \equiv \left\{ d \in (0, 0.498], F \in (max\{\overline{F}_{p2}^{vn}, \overline{F}_{p3}^{vn}\}, min\{\overline{F}_{s2}^{vn}, \overline{F}_{s3}^{vn}\}] \right\}$,

$\Omega_{AA}^{ss} \equiv \left\{ d \in (0, 0.554], F \in (max\{\overline{F}_{p2}^{ss}, \overline{F}_{p3}^{ss}\}, min\{\overline{F}_{s2}^{ss}, \overline{F}_{s3}^{ss}\}] \right\}$,

$\Omega_{RA}^{ps} \equiv \left\{ d \in (0, 0.466], F \in (max\{\overline{F}_{s3}^{ps}, \overline{F}_{p2}^{ps}\}, min\{\overline{F}_{s2}^{ps}, \overline{F}_{p3}^{ps}\}], \alpha \in (\overline{\alpha}_p^{ps}, \overline{\alpha}_s^{ps}] \right\}$,

$\Omega_{RA}^{vn} \equiv \left\{ d \in (0, 0.375], F \in (max\{\overline{F}_{s3}^{vn}, \overline{F}_{p2}^{vn}\}, min\{\overline{F}_{s2}^{vn}, \overline{F}_{p3}^{vn}\}], \alpha \in (\overline{\alpha}_p^{vn}, \overline{\alpha}_s^{vn}] \right\}$,

$\Omega_{RA}^{ss} \equiv \left\{ d \in (0.517, 0.554], F \in (max\{\overline{F}_{s3}^{ss}, \overline{F}_{p2}^{ss}\}, min\{\overline{F}_{p3}^{ss}, \overline{F}_{s2}^{ss}\}], \alpha \in (\overline{\alpha}_p^{ss}, \overline{\alpha}_s^{ss}] \right\}$,

$\Omega_{RR}^{ps} \equiv \left\{ d \in (0.543, 0.664), F \in (\overline{F}_{p2}^{ps}, \overline{F}_{s2}^{ps}] \right\}$, $\Omega_{RR}^{vn} \equiv \left\{ d \in (0.498, 0.664), F \in (\overline{F}_{s2}^{vn}, \overline{F}_{p2}^{vn}] \right\}$,

*and* $\Omega_{RR}^{ss} \equiv \left\{ d \in (0.554, 0.664), F \in (\overline{F}_{s2}^{ss}, \overline{F}_{p2}^{ss}] \right\}$.

Proposition 4 reflects that for any power structure, the selling mode equilibrium strategies of suppliers and platforms are consistent under certain conditions. Recalling Proposition 1 (iii), the supplier and platform can always choose the consistent selling mode when the cost of order fulfillment is moderate under the monopolistic supply chain

circumstance. We find that this conclusion is also true in the competitive supply chain circumstance. However, different from the monopolistic supply chain circumstance, in which the agency selling mode is the optimal strategy, under the competitive supply chain circumstance, the agency rate and competition intensity will also change the selling mode strategy that the two chains jointly prefer. Specifically, if the competition intensity is low, then both suppliers and platforms are willing to adopt the agency selling mode for cooperation, as shown in the blue region in Figure 6. The radical reason for this is that with the reduction in competition intensity, the demand and retail price of each chain gradually increase in the agency selling mode ($\partial p_j^{x,AA*}/\partial d < 0$, $\partial q_j^{x,AA*}/\partial d < 0$). In addition, when the competition intensity is low, and the agency rate is moderate, the equilibrium strategies of the players in the two supply chains are always opposite, as shown in the pink region in Figure 6. In other words, if the players in chain 1 prefer the reselling (agency selling) mode, then the players in chain 2 prefer the agency selling (reselling) mode. This is because the low competition intensity encourages supply chain players to choose the agency selling mode, while a moderate agency rate inhibits their motivation. Therefore, players of the two chains always tend to choose the opposite selling mode to optimize their own profits. Finally, when the competition intensity is high, the consistent equilibrium strategy of supply chain members is the scenario RR, as shown in the yellow region in Figure 6. We find that with the increase in competition intensity, the demand and retail price of the two chains show an upward trend ($\partial p_j^{x,RR*}/\partial d > 0$, $\partial q_j^{x,RR*}/\partial d > 0$). Therefore, high competition intensity always urges supply chain players to choose the reselling mode.

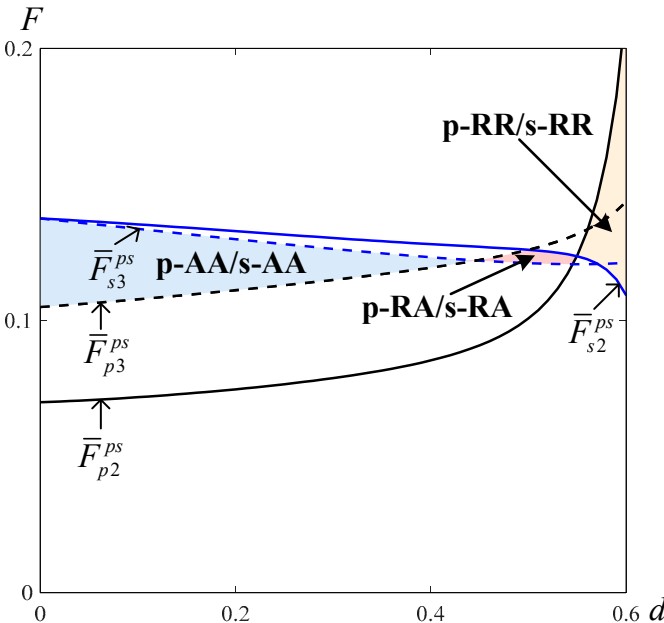

**Figure 6.** The suppliers' and the platforms' selling mode decisions under the competitive supply chain circumstance in the *ps* structure ($a = 1$, $\alpha = 0.55$).

Next, we discuss the selling mode equilibrium strategy of each total chain in the competitive environment (i.e., the equilibrium strategy of each chain after the players of each chain complete Pareto improvement). We define this situation as an integrated supply chain system:

**Proposition 5.** *Under the competitive supply chain circumstance, the equilibrium of the selling mode decision for the integrated supply chain system is expressed as follows:*

*(i) In the ps or vn power structure, scenario AA is the equilibrium decision if $d \in (0, 0.543]$ or $d \in (0, 0.498]$; otherwise, scenario RR is the equilibrium decision;*

*(ii) In the ss power structure, scenario AA is the equilibrium decision if $d \in (0, 0.517]$, and scenario RA or AR is the equilibrium decision if $d \in (0.517, 0.554]$; otherwise, scenario RR is the equilibirum decision.*

Proposition 5 reveals the equilibrium strategies of integrated supply chain systems under different power structures. According to Proposition 4 and Figure 7, the power structure and competition intensity are important factors affecting the selling mode equilibrium strategy of an integrated supply chain system. First, we find that when the leadership of the platforms is always not weaker than that of the suppliers, the two supply chains always tend to choose the same selling mode. In this case, if the competition intensity is low, then both chains opt for the agency selling mode, which is the equilibrium strategy. On the contrary, when the competition intensity is high, the reselling mode is the equilibrium strategy. This may be because with the increase in competition intensity, the total revenue of each chain gradually decreases in scenario *AA* ($\partial(\pi_{s,j}^{x,AA*} + \pi_{p,j}^{x,AA*})/\partial d < 0$) and gradually increases in scenario *RR* ($\partial(\pi_{s,j}^{x,RR*} + \pi_{p,j}^{x,RR*})/\partial d > 0$). In addition, when the platforms' leadership is always weaker than that of the suppliers, the two supply chains may prefer to choose the opposite selling mode when the competition intensity is moderate, as shown in Figure 7. This also reflects the influence of the power structure on the equilibrium decision of a competitive supply chain's selling mode. Moreover, we observe that the power structure affects the preference of each chain for the selling mode. First, the willingness of both chains to choose the agency selling mode is the strongest in the platform-Stackelberg structure and the weakest in the vertical Nash structure. Then, only in the supplier-Stackelberg structure are the two supply chains willing to opt for different selling modes. Finally, the willingness of both chains to choose the reselling mode is the strongest in the vertical Nash structure and the weakest in the supplier-Stackelberg structure.

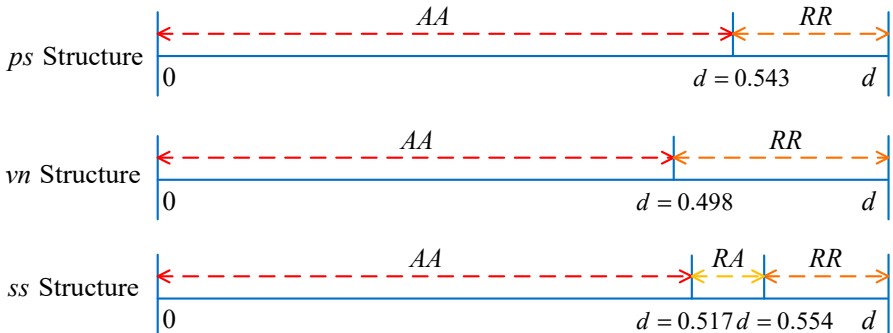

**Figure 7.** The selling mode decision for the integrated supply chain under the competitive supply chain circumstance.

It is noted that in the competitive supply chain circumstance, whether between platforms, suppliers, or the total chain, they all belong to both sides of the game, so a prisoner's dilemma may emerge. We investigated this question and show the results in Corollary 1:

**Corollary 1.** *(Prisoner's Dilemma). In the ps, vn, or ss power structures:*

*(i) Given $F \in (max\{\overline{F}_{p1}^{x}, \overline{F}_{p2}^{x}\}, \widetilde{F}_{p}^{x}]$, each platform choosing the agency selling mode (i.e., AA) results in a "prisoner's dilemma", whereas both platforms can obtain more payoffs with the reselling mode (i.e., RR) under such a circumstance;*
*(ii) Given $F \in (\widetilde{F}_{s}^{x}, min\{\overline{F}_{s1}^{x}, \overline{F}_{s2}^{x}\}]$, each supplier choosing the agency selling mode (i.e., AA) results in a "prisoner's dilemma", whereas both suppliers can obtain more payoffs with the reselling mode (i.e., RR) under such a circumstance;*
*(iii) Given $d \in (0.372, 0.543]$, $d \in (0.271, 0.498]$, or $d \in (0.372, 0.517]$, for the integrated supply chain, each supply chain choosing the agency selling mode (i.e., AA) results in a "prisoner's*

*dilemma", whereas both supply chains can obtain more payoffs with the reselling mode (i.e., RR) under such a circumstance.*

Corollary 1 shows that no matter what power structure, for platforms, suppliers, and the total supply chain, the phenomenon of the prisoner's dilemma will always appear under certain conditions. Specifically, (i) when the order fulfillment cost is moderate, the two platforms form a prisoner's dilemma, as shown in Figure 8a. At this time, for each platform, they always have more motivation to choose the agency selling mode, but this equilibrium strategy creates lower profits. Furthermore, by comparing the equilibrium profits of the two platforms under four different selling mode strategies, we find that each platform chooses the reselling mode to optimize their profits.

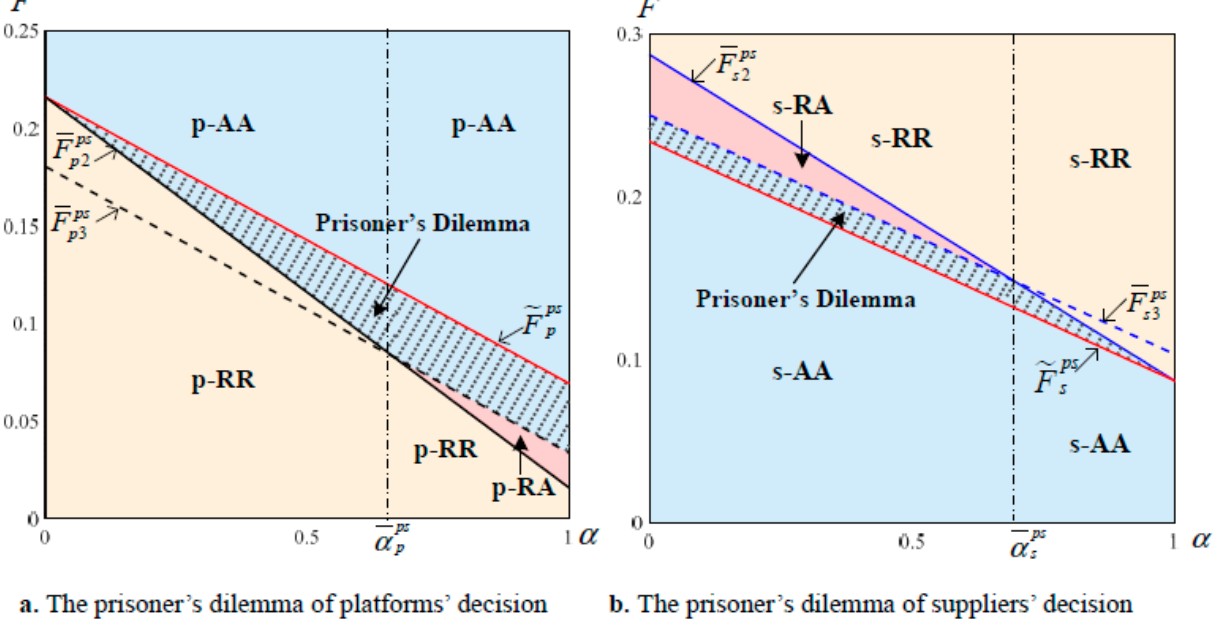

a. The prisoner's dilemma of platforms' decision     b. The prisoner's dilemma of suppliers' decision

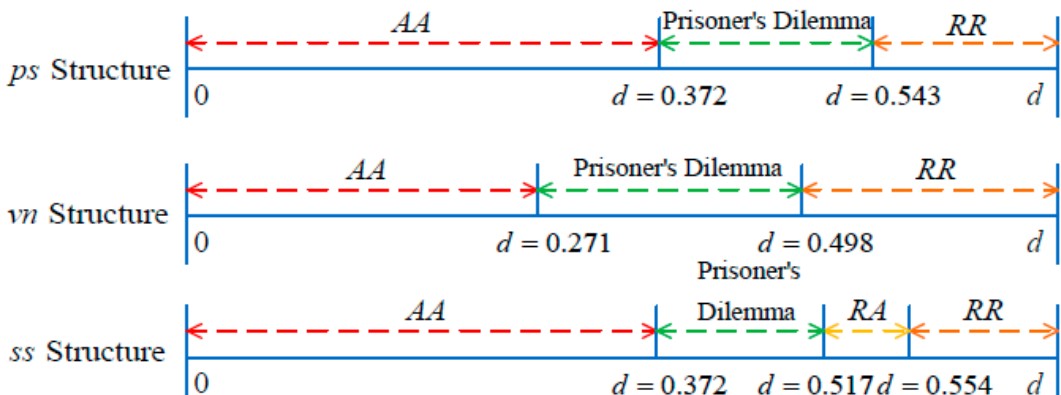

c. The prisoner's dilemma of the integrated supply chain system's selling mode strategies

**Figure 8.** The prisoner's dilemma of the selling mode strategies under the competitive supply chain circumstance ($a = 1$, $d = 0.45$). (**a**) The prisoner's dilemma of platforms' decision; (**b**) The prisoner's dilemma of suppliers' decision; (**c**)The prisoner's dilemma of the integrated supply chain system's selling mode strategies.

(ii) When the order fulfillment cost is moderate, the equilibrium strategy of the two suppliers leads to the prisoner's dilemma, as shown in Figure 8b. In this case, for each manufacturer, the moderate order fulfillment cost makes them always willing to choose the agency mode. However, for the two suppliers, this strategy does not optimize their

profits. Furthermore, by comparing the equilibrium profits of two suppliers under different strategies, we find that scenario *RR* is the equilibrium strategy to achieve the optimal profits for suppliers.

(iii) When the competition intensity is moderate, the equilibrium strategy of the two chains causes a prisoner's dilemma, as shown in Figure 8c. It is noted that, unlike the prisoner's dilemma region caused by the equilibrium decision of a single player, the intensity of competition is an important factor leading to the emergence of a prisoner's dilemma between the two chains. When the competition intensity is moderate, each chain has a strong willingness to choose the agency mode. However, we observed in Proposition 4 that with the increase in competition intensity, the profit of each chain in the agency mode gradually decreases. Therefore, for the two chains, the agency mode cannot make them achieve the optimal profit at this time. By comparing the profits of the two chains in four different scenarios, which are the same as the optimal strategy of a single player, we find that scenario *RR* makes each chain achieve their optimal profits.

Through Corollary 1, we know that under the competitive supply chain circumstance, it is not only difficult for the players in each supply chain to achieve consistency in their selling mode strategy, but they are also prone to a prisoner's dilemma between the two chains. In order to further explore the selling model equilibrium strategy to achieve the optimal profit of each chain in a competitive environment, we investigated the Pareto improvement of the whole supply chain system and defined it as a centralized supply chain system. The specific results are shown in Proposition 6:

**Proposition 6.** *Under the competitive supply chain circumstance, the equilibrium of the selling mode decision for the centralized supply chain system in the ps, vn, and ss power structure is as follows:*

*(i) Scenario AA is the equilibrium decision if $d \in (0, 0.349]$, $d \in (0, 0.263]$, or $d \in (0, 0.332]$;*
*(ii) Scenario RA or AR is the equilibrium decision if $d \in (0.349, 0.390]$, $d \in (0.263, 0.279]$, or $d \in (0.332, 0.417]$;*
*(iii) Otherwise, scenario RR is the equilibrium decision.*

Proposition 6 reveals that for any power structure, the whole supply chain system can achieve Pareto improvement. Moreover, the selling mode equilibrium strategy in the competitive supply chain circumstance is closely related to the intensity of the competition, as shown in Figure 9. Specifically, (i) when the competition intensity is low, it is the optimal strategy for the whole supply chain system to choose the agency selling mode for both chains. This is mainly because when the competition intensity is low, the profits of both supply chains are always optimal when choosing the agency selling mode. Furthermore, we can observe the differences in supply chain strategy selection in different power structures. The willingness of the supply chain system to choose the reselling mode is the highest in the power structure with the platform as the leader and the lowest in the vertical Nash structure. (ii) When the competition intensity is moderate, the optimal strategy of the whole supply chain system is to choose different selling modes for the two supply chains. Recalling Proposition 5, with the increase in competition intensity, the revenue of the supply chain gradually increases in the reselling mode and decreases in the agency selling mode. If the competition intensity is moderate, then it is difficult to achieve the optimization of the supply chain system if both chains only choose the reselling mode or agency selling mode. In addition, we find that the willingness of the supply chain system to choose scenario *RA* is the highest in the supplier-Stackelberg structure and the lowest in the vertical Nash structure. (iii) When the competition intensity is high, the supply chain system is optimal in the reselling mode selected by both supply chains. This is intuitive because the total profits of the two supply chains always increases with the increase in competition intensity. It is worth noting that we find that the preference for the supply

chain system for scenario *RR* is the highest in the vertical Nash structure and the lowest in the platform-Stackelberg structure.

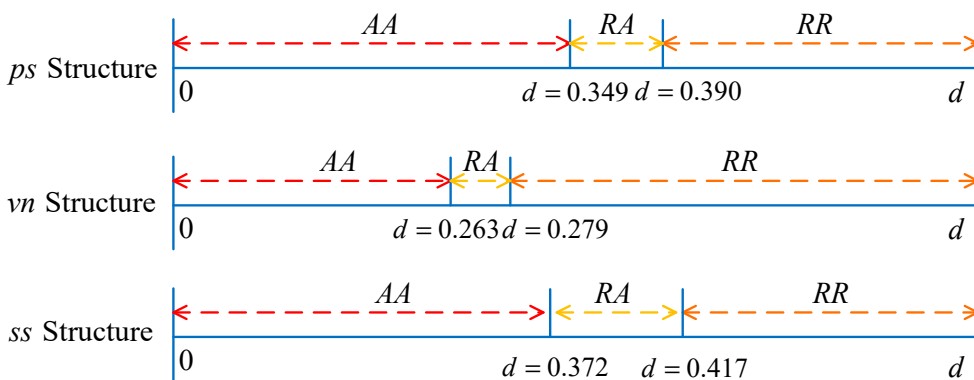

**Figure 9.** The selling mode decision for the centralized supply chain system under the competitive supply chain circumstance.

## 5. Comparative Analysis

Until now, we have analyzed the selling mode equilibrium strategy under monopolistic and competitive supply chain circumstances for each combination of power structures. In this section, we first compare the profits under monopolistic and competitive supply chains from the integrated or centralized supply chain system and further analyze the impact of different supply chain circumstances on the selling mode strategy. Then, we discuss the impact of the power structure on the selling mode strategy under monopolistic and competitive supply chains.

### 5.1. Comparison of Monopolistic and Competitive Supply Chains

In this subsection, we first derive the comparison results of a single supply chain's profit under the monopolistic supply chain circumstance and a single supply chain's profit under the competitive supply chain circumstance for each combination of power structures. Then, we go on to explore the comparison results between the complex competitive supply chain system's profit and the single monopoly supply chain system's profit:

**Proposition 7.** *In the ps, vn, or ss power structure, for any single supply chain, the competitive supply chain circumstance is better off than the monopolistic supply chain circumstance if $d \in (0.559, 0.664)$, $d \in (0.598, 0.664)$, or $d \in (0.559, 0.664)$ and is worse off otherwise.*

Proposition 7 investigates the optimal supply chain circumstance for any single supply chain in three different power structures. Interestingly, we find that for any single supply chain, the competitive supply chain circumstance can be more favorable than the monopolistic supply chain circumstance. When combining Propositions 2 and 7, it is not difficult to find that the supply chain under the competitive supply chain circumstance does not faithfully implement the agency selling model as the monopolistic supply chain circumstance does but instead chooses the reselling model when the competition is fierce. It is precisely because of this change that the single supply chain can be more profitable under competitive circumstances.

Furthermore, through the interval threshold of *d* under the different power structures, we find that the threshold of *d* under the platform-Stackelberg structure and the supplier-Stackelberg structure is lower than that of the vertical Nash structure, and thus under the competitive supply chain circumstance, for any single supply chain, the supplier-Stackelberg and platform-Stackelberg structures are more conducive power structures than the vertical Nash structure:

**Proposition 8.** *In the ps, vn, or ss power structure, the following applies for the centralized supply chain system:*

*(i) When $d \in (0, 0.349]$, $d \in (0, 0.263]$, or $d \in (0, 0.332]$, the competitive supply chain circumstance is better off than the monopolistic supply chain circumstance if $F \in (0, \overline{F}_{AA}^{ps}]$, $F \in (0, \overline{F}_{AA}^{vn}]$, or $F \in (0, \overline{F}_{AA}^{ss}]$ and is worse off otherwise;*
*(ii) When $d \in (0.349, 0.390]$, $d \in (0.263, 0.279]$, or $d \in (0.332, 0.417]$, the competitive supply chain circumstance is better off than the monopolistic supply chain circumstance if $F \in (0, \overline{F}_{RA}^{ps}]$, $F \in (0, \overline{F}_{RA}^{vn}]$, or $F \in (0, \overline{F}_{RA}^{ss}]$ and is worse off otherwise;*
*(iii) When $d \in (0.390, 0.664)$, $d \in (0.279, 0.664)$, or $d \in (0.417, 0.664)$, the competitive supply chain circumstance is better off than the monopolistic supply chain circumstance if $F \in (0, \overline{F}_{RR}^{ps}]$, $F \in (0, \overline{F}_{RR}^{vn}]$, or $F \in (0, \overline{F}_{RR}^{ss}]$ and is worse off otherwise. The expression of $\overline{F}_y^x$ is shown in the Supplementary Materials, $x \in \{ps, vn, ss\}$, and $y \in \{AA, RA, RR\}$.*

Proposition 8 explores the optimal supply chain circumstance for the centralized supply chain system in three different power structures. We find that the order fulfillment cost is an important factor affecting the quality of the supply chain circumstance. As shown in the example of the *ps* structure in Figure 10, if the order fulfillment cost is below a certain threshold, then the centralized supply chain system under the competitive circumstance is better off than the monopolistic circumstance, which is shown in the yellow region. On the contrary, if the order fulfillment cost is above the threshold, then the supply chain system under the monopolistic circumstance will perform more profitably, as shown in the blue region.

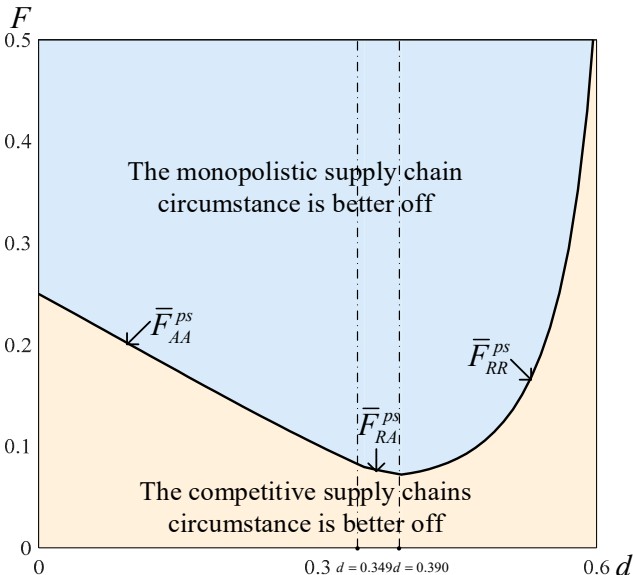

**Figure 10.** The comparison of monopolistic and competitive supply chain circumstances in the *ps* structure for the centralized supply chain system ($a = 1$, $\alpha = 0.55$).

**Corollary 2.** *In the ps, vn, or ss power structure, for the centralized supply chain system, when $d \in (0.390, 0.664)$, $d \in (0.279, 0.664)$, or $d \in (0.417, 0.664)$, the reselling mode (i.e., RR) will turn the competitive supply chain circumstance from worse off to better off than the monopolistic supply chain circumstance if $F \in (\overline{F}_{AA}^{ps}, \overline{F}_{RR}^{ps}]$, $F \in (\overline{F}_{AA}^{vn}, \overline{F}_{RR}^{vn}]$, or $F \in (\overline{F}_{AA}^{ss}, \overline{F}_{RR}^{ss}]$.*

Corollary 2 reveals that there exists the reverse region which turns the competitive supply chain circumstance from worse off to better off than the monopolistic supply chain circumstance. This expected reversal is achieved by changing the selling mode from the agency selling mode to the reselling mode under the competitive supply chain circumstance. We show an example of the *ps* structure in Figure 11. When the supply

chain's competition is fierce, the reselling mode makes the supply chain system more profitable under the competitive supply chain circumstance compared with the agency selling mode and thereby changes the competitive supply chain circumstance from worse off to better off compare with the monopolistic supply chain circumstance (see the red dotted region).

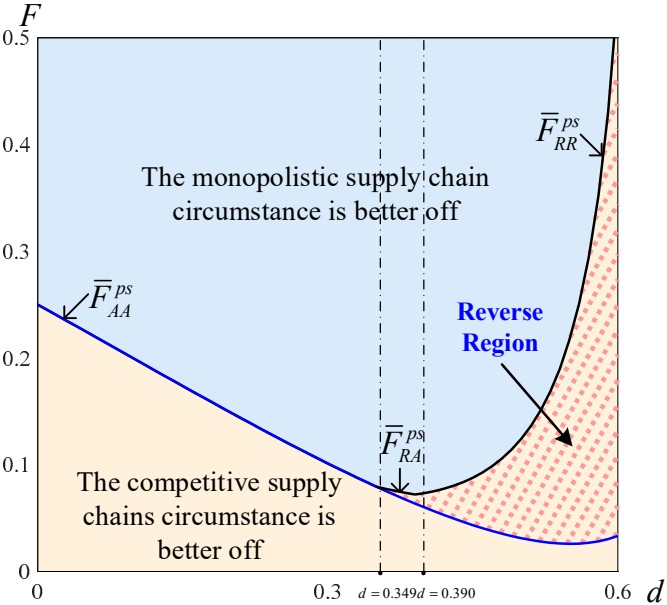

**Figure 11.** The impact of the selling mode of the centralized supply chain system under the monopolistic and competitive supply chain circumstances in the *ps* structure ($a = 1$, $\alpha = 0.55$).

By combining Propositions 7 and 8, we can find that introducing a competitive supply chain is not always bad for the single supply chain or the supply chain system. Under a highly competitive supply chain circumstance, changing the selling mode from the agency selling mode to the reselling mode under the competitive supply chain circumstance can make any supply chain or the centralized supply chain system under the competitive supply chain circumstance better than those under the monopolistic supply chain circumstance.

*5.2. Comparison of Power Structures*

In this subsection, we first analyze the impact of the power structure on players' selling mode strategies under monopolistic and competitive supply chain circumstances. Then, we explore the impact of the power structure on the comparison results between the complex competitive supply chain system's profit and the single monopoly supply chain system's profit:

**Proposition 9.** *The impact of the power structure on players' selling mode strategies is as follows:*

*(i) Under the monopolistic supply chain circumstance, $\overline{F}_{p1}^{ss} < \overline{F}_{p1}^{vn} < \overline{F}_{p1}^{ps}$ and $\overline{F}_{s1}^{ss} < \overline{F}_{s1}^{vn} < \overline{F}_{s1}^{ps}$;*
*(ii) Under the competitive supply chain circumstance, $\overline{F}_{p2}^{ss} < \overline{F}_{p2}^{vn} < \overline{F}_{p2}^{ps}$ and $\overline{F}_{p3}^{ss} < \overline{F}_{p3}^{vn} < \overline{F}_{p3}^{ps}$, while $\overline{F}_{s2}^{ss} < \overline{F}_{s2}^{vn} < \overline{F}_{s2}^{ps}$ and $\overline{F}_{s3}^{ss} < \overline{F}_{s3}^{vn} < \overline{F}_{s3}^{ps}$, where $\overline{\alpha}_{p}^{ss} < \overline{\alpha}_{p}^{vn} < \overline{\alpha}_{p}^{ps}$ and $\overline{\alpha}_{s}^{ss} < \overline{\alpha}_{s}^{vn} < \overline{\alpha}_{s}^{ps}$.*

Proposition 9 (i) shows the impact of the power structure on the player's selling mode strategy under the monopolistic supply chain circumstance. For both the platform and the supplier, the order fulfillment cost thresholds are the highest under the *ps* structure and the lowest under the ss structure. This means that both the platform and the supplier prefer to play the "reseller" role under the *ps* structure more, which is the platform more willing to choose the reselling mode, and the supplier prefers the agency selling mode under the *ps* structure more. From Proposition 9 (ii), we can derive how the power structure

affects the player's selling mode strategy under the competitive supply chain circumstance. Similar to the monopolistic supply chain circumstance, both of the order fulfillment cost thresholds are the highest under the *ps* structure and the lowest under the ss structure for the platform and the supplier, and thus both platforms prefer the reselling strategy more, and both suppliers are more inclined to choose the agency selling mode under the *ps* structure. Furthermore, we also derive that the platform fee rate threshold is the highest under the *ps* structure and the lowest under the ss structure. By combining Proposition 3 and Lemma 1, we can find that under the competitive supply chain circumstance, both platforms are more able to achieve a consistent selling mode under the *ps* structure, and the suppliers are more able to achieve a consistent selling mode under the ss structure:

**Proposition 10.** *The impact of the power structure on the comparison of monopolistic and competitive supply chain circumstances is expressed as follows: (i)* $\overline{F}_{AA}^{ss} = \overline{F}_{AA}^{ps} = \overline{F}_{AA}^{vn}$, *(ii)* $\overline{F}_{RA}^{ps} < \overline{F}_{RA}^{ss} < \overline{F}_{RA}^{vn}$ *when* $d \in (0, 0.392]$; *otherwise,* $\overline{F}_{RA}^{ss} = \overline{F}_{RA}^{ps} > \overline{F}_{RA}^{vn}$, *and (iii)* $\overline{F}_{RR}^{ss} = \overline{F}_{RR}^{ps} < \overline{F}_{RR}^{vn}$ *when* $d \in (0, 0.471]$; *otherwise,* $\overline{F}_{RR}^{ss} = \overline{F}_{RR}^{ps} > \overline{F}_{RR}^{vn}$.

Proposition 10 indicates how the power structure affects the comparison results of the supply chain system under monopolistic and competitive supply chain circumstances. From Proposition 10 (i), when both supply chains choose the agency selling mode (i.e., *AA*) as the optimal selling mode under the competitive circumstance, the power structure has no effect on the results. However, part (ii) shows that when the supply chain chooses the opposite selling mode (i.e., *RA*), the order fulfillment threshold is the highest under the *vn* structure when the supply chain's competition is moderate, while the threshold is the highest under the *ps* or *ss* structure when the competition is fierce. Similar to part (ii), part (iii) shows that when both supply chains choose the reselling mode (i.e., *RR*), the order fulfillment threshold is the highest under the *vn* structure when the supply chain's competition is moderate, while the threshold is the highest under the *ps* or *ss* structure when the competition is fierce. By combining Proposition 8 and the above results, we conclude that under the moderately competitive supply chain circumstance, the *vn* structure is the most favorable power structure, and under the fiercely competitive circumstance, the *ps* and *ss* structures are the more favorable power structures.

## 6. Concluding Remarks

### 6.1. Conclusions

In this paper, we constructed a game theoretic model from the supplier-Stackelberg, vertical Nash, and platform-Stackelberg game perspectives under monopolistic and competitive circumstances to explore the impact of competition and the power structure on the strategy making of a supply chain's selling mode. In each power structure, based on the selling mode selection strategy of the supply chain, five scenarios were investigated in the framework. By comparing the optimal profits of the players, we found the equilibrium strategy of the selling mode of supply chains under the monopolistic or competitive circumstances. The main findings are as follows.

First, we investigated the influence of a monopoly or competition on the strategy making of a supply chain's selling mode. For the platform and the supplier, their selling mode strategies depend on the order fulfillment cost coefficient in the monopolistic circumstance while depending not only on the order fulfillment cost coefficient but the platform fee in the competitive circumstance, and we found that the supplier and the platform had conflict over the selling mode strategy in the vast majority of cases.

Next, through Pareto improvement, the players can achieve coordination and promote the players' payoff improvement in both monopolistic and competitive circumstances. The results of Pareto improvement show that the agency selling mode is always the optimal choice in the monopolistic circumstance, while the intensity of competition is an important factor that affects the selling mode strategy of each chain and the whole supply chain system in the competitive circumstance. Specifically, when the competition intensity is low,

the agency selling mode is always an equilibrium strategy, while when the competition intensity is high, the reselling mode is the optimal strategy. Furthermore, it is worth noting that each player choosing the agency selling mode will result in a "prisoner's dilemma" in the competitive circumstance, where both players can obtain more payoffs with the reselling mode, and such a "prisoner's dilemma" cannot be eliminated by Pareto improvement.

Lastly, we compared the payoffs of the single supply chain in the monopolistic and competitive circumstances and derived that the single supply chain in the monopolistic circumstance often gains more profits than in the competitive circumstance, except for the specific case of adopting the reselling mode when the competitive circumstance is fierce. In addition, we found the impact of the power structure on the players' selling mode strategies and the results of Pareto improvement in the monopolistic and competitive circumstances. We found that the willingness of platforms and suppliers to play the role of "reseller" is the strongest under the *ps* structure and the weakest in the *ss* structure, whether in the monopolistic and competitive circumstances. In addition, according to the results for Pareto improvement, the supply chain is most inclined to the agency selling mode under the *ps* structure and least inclined under the *vn* structure in the competitive circumstance.

### 6.2. Managerial Insights

Based on the main findings, we provide the following valuable guidance for managers. First, we innovatively consider the equilibrium strategies of the members' selling mode in the monopolistic and competitive supply chain circumstances, providing decision guidance for the supply chain parties in different circumstances. In addition, by comparing the monopolistic and competitive supply chain circumstances, we found that if the supply chain changes the selling mode from the agency selling mode to the reselling mode, then the profit of the supply chain or the whole supply chain system will be better in a competitive circumstance (e.g., Huawei, which is in the highly competitive mobile phone industry, cooperates with JD.com through the reselling mode in order to maintain high profits). This also shows that by adjusting the selling mode of the members, the supply chain can reduce the loss caused by competition with the supply chain, making the competitive circumstance better than the monopolistic circumstance. Third, we found that the power structure has a profound impact on the members' selling mode decisions and profits. Under the competitive supply chain circumstance, the whole supply chain system is more willing to choose the agency selling mode in the *ps* structure than the other two power structures (e.g., the clothing brand gap has a weak leadership position relative to JD.com, so it has reached an agency selling agreement with JD.com). In addition, in the competitive supply chain circumstance, if the competition is mild, then the *vn* structure is the most favorable power structure of the whole supply chain; otherwise, the opposite is true. Finally, we verified that parties will maximize their own benefits according to rationality, and their decisions are not always consistent. However, whether in monopolistic or competition circumstances, parties can achieve the optimization of a supply chain through Pareto improvement.

### 6.3. Limitations and Future Research Directions

Although this paper comprehensively examined the interaction between competition, power structure, and selling mode decision, our paper also has several limitations. First, we studied the performance of two supply chains in three power structures in a competitive circumstance and assumed that the two chains had the same power structure. However, the power structure of each supply chain may not be consistent, which can be discussed in future research. Second, suppliers gradually began to sell products through a combination of the two selling modes and platforms. For example, JD.com has established its own stores of Xiaomi and Oppo in its own mall, while Xiaomi and Oppo have also opened their own flagship stores on JD.com to sell products to consumers. Therefore, research on the dual-channel sales composed of the reselling and agency selling modes in the platform supply chain is also interesting. Third, we assumed that the information was completely symmetrical, but the platform that masters massive sales data may have more market

demand information. Therefore, the impact of information can be further explored. Finally, this paper used game theory to study a two-level supply chain. In the future, we can expand this research and develop decision support systems (e.g., decision trees) to discuss n-level supply chains.

**Supplementary Materials:** The following supporting information can be downloaded at: https://www.mdpi.com/article/10.3390/su141711016/s1, Table S1: Outcomes in scenario R and A. Table S2: Outcomes in scenario RR. Table S3: Outcomes in scenario RA. Table S4: Outcomes in scenario AA. All listings and proofs: Listing S1: Thresholds in Proposition 3. Listing S2: Thresholds in Lemma 1. Listing S3: Thresholds in Proposition 8. Proof of Table S1. Proof of Proposition 1. Proof of Proposition 2. Proof of Table S2. Proof of Table S3. Proof of Table S4. Proof of Proposition 3. Proof of Lemma 1. Proof of Proposition 4. Proof of Proposition 5. Proof of Corollary 1. Proof of Proposition 6. Proof of Proposition 7. Proof of Proposition 8. Proof of Corollary 2. Proof of Proposition 9.

**Author Contributions:** Conceptualization, T.F. and L.Z. (Lihao Zhang); methodology, L.Z. (Lixi Zhou); software, J.Y.; validation, L.Z. (Lixi Zhou); formal analysis, L.Z. (Lixi Zhou), J.Y. and L.Z. (Lihao Zhang); investigation, L.Z. (Lixi Zhou), T.F. and J.Y.; resources, L.Z. (Lixi Zhou) and T.F.; writing—original draft preparation, L.Z. (Lixi Zhou) and J.Y.; writing—review and editing, T.F. and L.Z. (Lihao Zhang); visualization, L.Z. (Lixi Zhou), J.Y.; supervision, T.F. and L.Z. (Lihao Zhang); funding acquisition, T.F. and L.Z. (Lihao Zhang). All authors have read and agreed to the published version of the manuscript.

**Funding:** National Natural Science Foundation of China: 72032001, 71972071, 71971137; Scientific and Innovative Action Plan of Shanghai: 21692109300, 22692110700; Chenguang Program of Shanghai Municipal Education Commission: 20CG56.

**Conflicts of Interest:** The authors declare no conflict of interest.

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
