# Peer review of "Monopolistic vs. Competitive Supply Chain Concerning Selection of the Platform Selling Mode in Three Power Structures"

_sustainability, doi:10.3390/su141711016_

Round 1

Reviewer 1 Report

1. Introduction:

The introduction is exhaustive and consistent, but authors did not make any reference to the stream of literature dedicated to the Behavioral Operation Management. I think that they must introduce references and recent contributions about this topic like:

- Pournader, M., Ghaderi, H., Hassanzadegan, A., & Fahimnia, B. (2021). Artificial intelligence applications in supply chain management. International Journal of Production Economics, 241, 108250.

-Vipin, B., and R. K. Amit. "Wholesale price versus buyback: A comparison of contracts in a supply chain with a behavioral retailer." Computers & Industrial Engineering 162 (2021): 107689.

-Chaudhuri, Atanu, Manjot Singh Bhatia, Yasanur Kayikci, Kiran J. Fernandes, and Samuel Fosso-Wamba. "Improving social sustainability and reducing supply chain risks through blockchain implementation: role of outcome and behavioural mechanisms." Annals of Operations Research (2021): 1-33.

-D’Urso, Diego, Ferdinando Chiacchio, and Evangelia Demerouti. "Measuring How Decision Support Systems Improve Newsvendors’ Performance: The Subjects’ Version." Sustainability 13, no. 18 (2021): 10251.

This can be also be done in the Literature Review

2. Literature Review

a) The font is smaller than section 1

b) As said in the previous comment, authors should also find references that discuss the supply chain from the point of view of the behavioural operations management where many biases in the actions of the supply chain actors are reported and discussed according to the setting of the supply chain echelons. I just gave few examples of manuscripts that introduce different aspect from several point of views. Please add these references at least

c)The summary Table 1 is interesting because it shows where the research paper is positioned with respect to similar ones. Nevertheless, authors should also add what are the hypotheses and limitations of this research.

3. The model

a) Line 203, there is a semicolumn and a capital letter afterwards. Please correct it.

b) The same error is in row 236. Please correct throughout the paper.

c) Do authors think that the model can be generalized for a n-level supply chain or not? They should extend this concept in the model assumptions.

d) As I noticed, the assumptions of the models are spreaded in the section 3. Maybe a resuming table can help the readers to have a more clear understanding of the hypotheses used in the model.

4 - 6: The core of the paper, the experimental part is very comphrensive but:

a) the Sections 4 and 5 should be organized in order to facilitate the reading. First of all I would collapse the section in one section (4) that, is obviously organized in two subsections, Monopolistic (with the assumptions and its scenarios) and the competitive. Authors should add a general paragraph that explain why they can focus on these two main competitive scenarios and give some short information about them. 

a) Line 485: Intuitively, the selling mode equilibrium strategies of the platforms and suppliers are quite inconformity -> ...are quite different

The noun "inconformity" is not correct, I guess.

7. Conclusions are satisfactory

Reviewer 2 Report

The authors study a platform supply chain and applied a game theory to model a monopolistic and a competitive circumstance with consideration of three power structures. The model is realistic and the analysis is solid. The manuscript is well written and the literatures are adequate. I suggest that this paper could be accepted for publication after some minor revision.

 1) Figures 1 and 2 demonstrate the monopolistic and competitive supply chain structures respectively, yet the necessary instructions about the lines in these figures are missing. Moreover, Figure 7 should give the values of related parameter.

 2) In Tables 3~6, the upper left corner of the table needs to explain the relevant parameters and the scenarios. Moreover, I suggest that these tables of optimal solutions should be shown in the Appendix, which can make the paper more concise and does not affect the narrative.

 3) In Section 7, the managerial insights obtained from the research should be highlighted, and should be given in a sub-section. Similar point that need to be highlighted is limitations of this research.

 4) In the literature review, the format of several literature is inconsistent with the general, such as, [2], [40], etc.

Round 2

Reviewer 1 Report

-

Author Response

We are grateful to Reviewer 1 who has found our first revision satisfactory.